

**Editorial: The shadowlands of science communication in academia — definitions, problems, and possible solutions**

Shahzad Gani[1,2], Louise Arnal[3*], Lucy Beattie[4], John Hillier[5], Sam Illingworth[6], Tiziana Lanza[7], Solmaz Mohadjer[8], Karoliina Pulkkinen[9], Heidi Roop[10], Iain Stewart[11], Kirsten von Elverfeldt[12], Stephanie Zihms[13]

[1]Centre for Atmospheric Sciences, Indian Institute of Technology Delhi, New Delhi, India
[2]Institute for Atmospheric and Earth System Research/Physics, University of Helsinki, Helsinki, Finland
[3]Centre for Hydrology, University of Saskatchewan, Canmore, Alberta, Canada
[4]School of Education and Social Sciences, University of the West of Scotland, Scotland
[5]Department of Geography and Environment, Loughborough University, Loughborough, UK
[6]Department of Learning and Teaching Enhancement, Edinburgh Napier University, Edinburgh, Scotland
[7]Istituto Nazionale di Geofisica e Vulcanologia, Rome, Italy
[8]Global Awareness Education, University of Tübingen, Tübingen, Germany
[9]Aleksanteri Institute, University of Helsinki, Helsinki, Finland
[10]University of Minnesota Climate Adaptation Partnership, St Paul, Minnesota, USA
[11]Royal Scientific Society, Amman, Jordan
[12]Department of Geography and Regional Studies, Alpen-Adria-Universität Klagenfurt, Klagenfurt, Austria
[13]Academic Writing Centre and Graduate School, Glasgow Caledonian University
*Now at Ouranos, Montreal, Quebec, Canada

*Correspondence to*: Shahzad Gani (shahzadgani@iitd.ac.in)

**Abstract.**

Science communication is an important part of research, including in the geosciences, as it can benefit society,
science, and make science more publicly accountable. However, much of this work takes place in "shadowlands" that are neither fully seen nor understood. These shadowlands are spaces, aspects, and practices of science communication which are not clearly defined and may be harmful with respect to the science being communicated or for the science communicators themselves. With the increasing expectation in academia that researchers should participate in science communication, there is a need to address some of the major issues that lurk in these
shadowlands. Here the editorial team of *Geoscience Communication* seeks to shine a light on the shadowlands of geoscience communication and suggest some solutions and examples of effective practice. The issues broadly fall under three categories: 1) harmful or unclear objectives; 2) poor quality and lack of rigor; and 3) exploitation of science communicators working within academia. Ameliorating these will require: 1) clarifying objectives and audiences; 2) adequately training science communicators; and 3) giving science communication equivalent
recognition to other professional activities. By shining a light on the shadowlands of science communication in academia and proposing potential remedies, our aim is to cultivate a more transparent and responsible landscape for geoscience communication—a transformation that will ultimately benefit the progress of science, the welfare of scientists, and more broadly society at large.




# 1 Introduction: Science communication and geosciences

## 1.1 Defining science communication

Science communication is a broad field that has been growing and evolving over the last few decades. At the start
of this century, its remit and scope had expanded, with Burns et al. (2003, p. 183) recasting it as "the use of appropriate skills, media, activities, and dialogue to produce one or more of the following personal responses to science: Awareness, Enjoyment, Interest, Opinion-forming, and Understanding." Since then, over the following two decades, the theory and practice of science communication has continued to broaden, drawing in an ever-wider set of different actors and disciplines. As a result, this definition appears limited and outdated now.

In the 1980s, the initial motivation behind the Public Understanding of Science (PUS) movements was what became known as the "deficit model". This model operated under the assumption that the public's skepticism towards modern science was caused by a lack (deficit) of scientific knowledge implying that the public receives information passively. The belief was that scientists should put more effort to convey information to the public, and that as a
result people would change their opinions and develop a positive attitude towards science. However, it is now understood that the public communication of science is far more complex than what the knowledge deficit model suggests. Although the discredited deficit model continues to persist in scientific circles (Cortassa, 2016; Simis et al., 2016), even among the core practitioners of the deficit model there is an acceptance of the need to reconsider science communication in light of a deeper understanding of contemporary society. While most practioners of the
deficit model agree with Fischhoff and Scheufele (2013) that communication is a two-way process, wherein scientists must both listen and speak, they also argue that this process should adhere to the same rigorous standards of evidence as the science itself. They make a case for science communication that is grounded in existing scientific research and subjected to empirical evaluation, rather than relying solely on intuition. Others, notably Bucchi and Trench (2021), argue that viewing science communication as a social conversation expands the concept of quality
beyond mere impact or effectiveness. This perspective encourages a multifaceted understanding of quality, where the evaluation of the quality of the conversation should not be based solely on the assessment of one participating party.

Bucchi and Trench (2021) also highlight the importance of acknowledging the philosophical background of science
communication. They emphasise that the distinction between the subject-researcher and the object-nature does not apply to the study of society, language, and culture, where the researcher is deeply intertwined within the subject matter. Furthermore, in redefining science communication as a "social conversation around science", Bucchi and Trench (2021) recognise public communication of science as an ongoing process rather than an end product. There is now a wealth of illustrative instances showcasing the use of diverse formats, such as performance, music, and
visual arts, to engage diverse publics with a wide range of scientific topics (Parks and White, 2021; Zaelzer, 2020; Lesen et al., 2016). The emergence of art-science has prompted a broader and non-prescriptive approach to science communication that encompasses various languages and formats, encouraging public engagement and discussion about science's role in society.

The range of tools employed in the field is diverse, spanning from science journalism and institutional communication through social media to public relations and marketing. It extends further to encompass museum





exhibitions, science events organised by cities and countries in collaboration with marketing and event management firms, science centers, science cafés, science slams, science blogs, and more. Weingart and Guenther (2016) add that even the traditional role of providing scientific advice to policymakers has been rebranded as science

communication. There are now multiple and even contested definitions of science communication. Weingart and Guenther (2016) highlight that science communication has evolved into an industry over the past few decades. It is no longer solely undertaken by a few dedicated scientists, science journalists, or popularizers with the intention of informing an interested public about the latest research advancements and their broader societal implications. Instead, science communication has become a battleground where various stakeholders compete for attention,

power, and influence due to financial interests, job opportunities, and professional identities. Consequently, even the definition of science communication itself is subject to debate and contention. Given this plurality in definitions and practices, it is important to acknowledge the spectrum of science communication and communicators.

For the purpose of this editorial and the *Geoscience Communication* (*GC*) journal, we refer to Hillier et al. (2021,

p-494) for a working definition of science communication: "We use the term "geoscience communication" to refer to the range of activities included in *GC*; these fall within a spectrum. At one end is activity-led work that might variously be known as education, outreach, communication, or engagement (e.g. science theatre as a medium for effective dialogue), and at the other end is curiosity-led research (e.g. how video games tangentially communicate geoscientific concepts) into how people engage with geoscience."

GC engages with science communication and communicators in five broad areas (Illingworth et al., 2018), illustrated by recent and interesting *GC* articles that embody these areas:
- *Geoscience education*: McGowan et al. (2022) explore the potential for using video games as a tool for teaching geoscience, specifically the geology and geomorphology of Hokkaido, Japan.

- *Geoscience engagement*: Fonseca et al. (2022) focus on the way physical concepts like the jet stream are represented in the press
- *Geoscience policy*: Brimicombe et al. (2022) investigate the bias of reporting various climate risks in English-language news articles.
- *History and philosophy of geosciences*: Rogers et al. (2022) examine the need for decolonizing the

curriculum for geologists.
- *Open geosciences*: Watson et al. (2023) evaluated the dissemination of satellite-based ground deformation measurements through Twitter.

Together, these recent *GC* articles demonstrate the diverse and multifaceted nature of geoscience communication.

**1.2 Importance of (geo)science communication**

Science communication is regarded important for many reasons, and there is no better way to demonstrate its multiple motivations than surveying the different goals that are regarded as central to science communication

(Table 1). The many goals listed by Kappel and Holmen (2019) and Besley et al. (2018) may be viewed as indicative of three broader values attached to science communication: (i) benefitting society, (ii) benefitting science, and (iii) making science more publicly accountable.

**Table 1:** Taxonomy and goals of science communication based on literature. Each of these goals is also linked to

one or more of the following broader values: (i) benefiting society, (ii) benefiting science, and (iii) making



science more publicly accountable. This is a rough categorization, as each of the goals may link to each of the three values.

| **Kappel and Holmen (2019)** introduce a taxonomy of the aims of science communication | **Besley et al. (2018)** list two baseline and six additional objectives related to a multi-dimensional understanding of trust, fairness, and the importance of identity |
|---|---|
| Improving the population's beliefs about science (i, ii) | Ensuring people are informed about scientific issues (i, iii) |
| Generating social acceptance (i, ii, iii) | Getting people interested or excited about science (iii) |
| Generating public epistemic and moral trust (i, ii, iii) | Demonstrating the scientific community's expertise (ii, iii) |
| Collect citizens' input about acceptable/worthwhile research aims and applications of science (ii, iii) | Hearing what others think about scientific issues (i, iii) |
| Generating political support for science (i, ii) | Demonstrating that the scientific community cares about society's well-being (iii) |
| Collect and make use of local knowledge (i, ii) | Demonstrating the scientific community's openness and transparency (iii) |
| Make use of distributed knowledge or cognitive resources to be found in the citizenry (i, ii) | Demonstrating the scientists share community values (iii) |
| Enhance the democratic legitimacy of funding, governance and application of science or specific segments of science (i, iii) | Framing research implications so members of the public think about a topic in way that resonates with their values (i, iii) |

The first value of science communication – benefitting society – arises from the prevalent idea that science communication is to improve a population's belief in science (Kappel and Holman, 2019), ensuring that people are informed about science, or interested or excited about scientific issues (Besley et al., 2018). The broader value of benefitting science as an institution – value (ii) – include goals such as generating public trust, collecting citizens input about acceptable research aims and applications of science, and generating political support for science,
amongst other examples (Kappel and Holman 2018). Aside from clear benefits for science, the goals listed by Besley et al. (2018) emphasise the public accountability of science – value (iii) –  by highlighting goals such as showing that the scientific community cares about society's well-being, demonstrating openness and transparency; demonstrating that scientists share community's values; framing research implications so members of the public think about a topic in way that resonates with their values.

Geoscientists, specifically, are working on many topics which can be directly relevant for the wellbeing of humans and other species. According to Cross and Congreve (2021), in order to tackle "wicked problems" such as climate change, it is vital for academics to possess higher level skills in communication, in addition to their domain-specific technical skillsets. They argue that as educators to undergraduate students and early career researchers, it is the duty of Geoscience academics to develop these skills. Oreskes (2020) makes a moral case for scientists to alert
society about threats that ordinary people have no other way of finding about. However, she also cautions "expertise is by definition specific, and so the obligation to speak up in our areas of expertise implies a reciprocal obligation to respect the expertise of others. Put another way: we have obligations both to speak and to listen. We need to speak up, to act as sentinels, and to be witnessing professionals in our domain of expertise, but we also need to act



with respect for colleagues who are the appropriate witnessing professionals in other domains" (Oreskes, 2020, p-43).

Multiple surveys across different regions show the high level of trust public has for scientists, especially university scientists (Krause et al., 2019). This puts scientists/academics in a unique position as science communicators. Because people listen to and trust scientists, they also expect scientists to disclose sensitive information (Thompson et al., 2023). Scientists, knowing they are in a unique position, also feel responsible and obliged (to what extent they are morally responsible is the question) for sharing sensitive information with the public. Since the communication channels between academics and the public are diverse, it is also expected that academics handle these channels with great care, making sure what they know and what they don't know (e.g., uncertainties) are clearly acknowledged and explained. Often academics are expected by the public to have all the answers (e.g., the case of the COVID-19 pandemic) — and not to make mistakes. This requires scientists to be clear, effective, and honest communicators, but perhaps more importantly, it requires them to be kind, empathetic, and humble.

### 1.3 Role of the *Geoscience Communication* journal and purpose of this editorial

Science communication is a crucial component of research, particularly within the field of geosciences, as it can benefit society, advance scientific understanding, and make science more publicly accountable. Launched in April 2018, *GC* is an international, interdisciplinary journal for articles on geoscience education, geoscience engagement, geoscience policy, history and philosophy of geosciences, open geoscience, and citizen science. *GC* provides a supportive platform for geoscientists, educators, and communicators to share their innovative communication approaches. The core purpose of *GC* is 2-fold (Illingworth et al., 2018): (1) to provide wider and more formal recognition for existing and future geoscience communication initiatives, and (2) to better formalise the discipline of geoscience communication.

In line with the core purpose of *GC*, in this editorial we highlight systemic issues ingrained in science communication, especially as it relates to the geosciences and geoscientists in academia. We refer to these issues as "shadowlands" hereafter. We also discuss the divergent perspectives and the spectrum of viewpoints among the authors of this editorial to mirror to some extent the spectrum of perspectives within the wider community. Finally, we propose potential solutions for the identified problems, and establish the journal's guiding principles.




## 2 The shadowlands of science communication

A lot of science communication in academia happens in "shadowlands", i.e., spaces, aspects, and practices which are not clearly defined and may be harmful with respect to the science being communicated or for the science communicators themselves. We outline three such shadowlands of science communication in academia in this article: 1) potentially harmful objectives, 2) poor quality and lack of rigor, and 3) exploitation of science communicators. We would like to point out that, as the authors of this editorial, we do not share the same views on all topics discussed herein. Our opinions span a broad spectrum, some of which are illustrated in Figure 1.

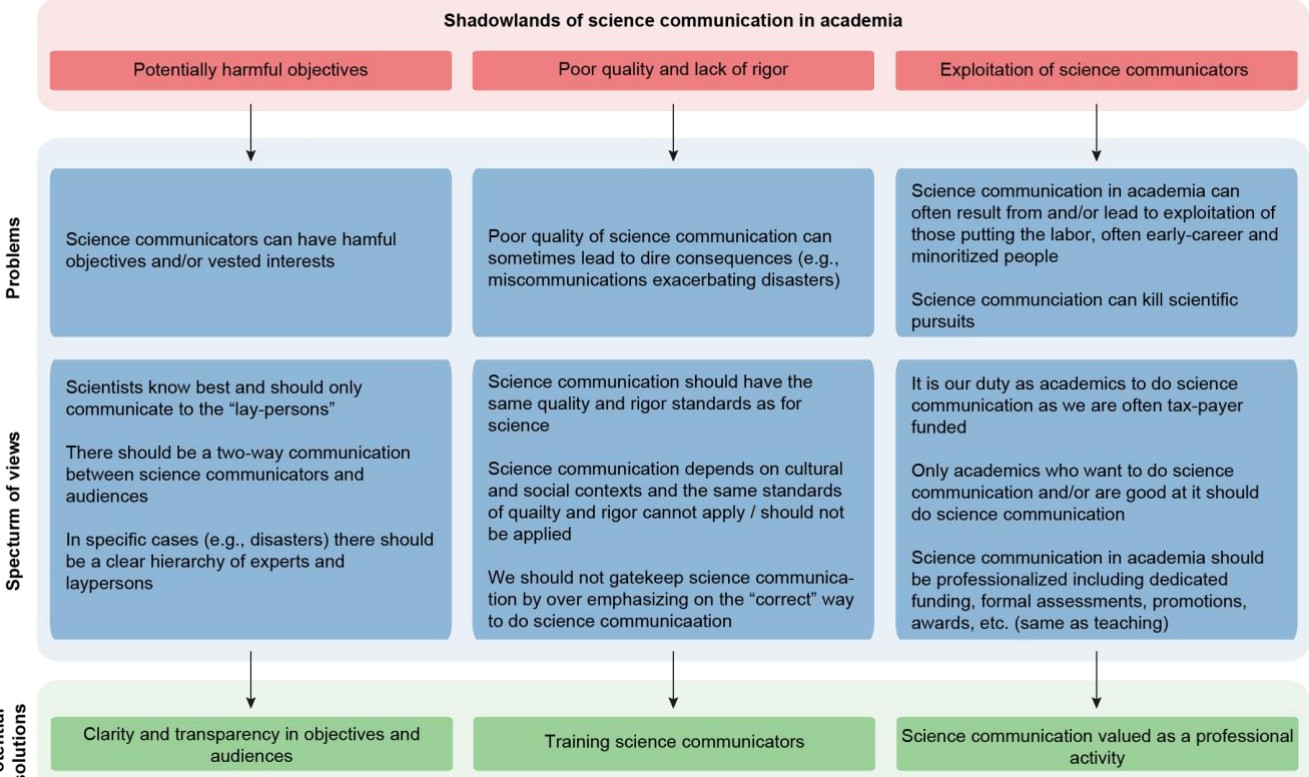

**Figure 1:** The shadowlands of science communication in academia — problems, spectrum of views, and potential solutions.

## 2.1 Potentially harmful objectives of science communication

While science communication is generally seen as a good (i.e., morally right) thing to do, there are some valid concerns about the objectives of the science communication, especially as they relate to the motivations of the science communicators. One major concern is the influence funders (when they exist) have on science communicators and science communication and their potential vested interests. As well as the ethical dimensions of science communication, what is the basic purpose of the science communicator? What are the terms on which science gets "made and sold"? How do we manage the powerful persuasive tool of storyifying science? Is success judged on whether we influence, persuade, and modify perceptions and behaviours? While there may not be one





"correct" answer to these fundamental questions about the purpose of science communication, reflecting on these can help us identify unintentional internal factors and hidden external factors that can lead to harmful science communication.

The multiple goals of science communication (Table 1) raise the concern of potential tension between different aims. This could be the case when the concerns raised by the public differ from scientists' own evaluation of what is best for society's well-being. Resolving such tensions can be difficult; the public's views can be based on serious misconceptions, but prioritising scientists' own conceptions (positionality) of societal well-being can risk being paternalistic. Aside from the issue of tension between many aims, there is also the worry that the goals of
professional science communication might conflict with the core aims or norms of the relevant scientific disciplines. For example, some scientific discipline may draw especially careful conclusions on the basis of their data, but such nuances might not lend themselves for "punchy" storytelling preferred in the media. This concern raises its head especially when professionalization of science communication means that "there is money in the game, there are jobs to be captured, and there are professional identities at stake." (Weingart and Guenther, 2016,
p-2). Another instance of tension between goals of science communication and the core disciplinary goals relates to "marketing-led" science communication, which can be incompatible with promoting long-term sustainability (Stewart and Hurth, 2021).

Aside from such instances of potential tension, there is also the question of due process — especially regarding the
model of communication and valuable attributes of communication. A major challenge with the broader goal of "informing the public" concerns the so-called deficit model, where the public is viewed as having insufficient knowledge of science which is remedied by scientists' successful communication. Although issues related to the deficit model of science communication are well known (see e.g., Sturgis and Allum 2004) it is still regarded a viable model for influencing science policy (Cortassa, 2016; Simis et al., 2016) and there is evidence that scientists
endorse it (Besley and Nisbet 2013). With respect to communicative virtues, openness, honesty, and transparency in science communication are usually recommended (e.g., Wilsdon and Willis, 2004; Keohane et al., 2014). However, there have been some concerns raised that exercising these virtues in science communication can undermine public trust in science (John, 2018). The notion of the 'zombie' deficit model is important to note, but equally we should acknowledge that one-way awareness raising mechanisms occasionally have their place, e.g. in
emergency risk communication situations where actionable risk messaging is required. In such situations, the emphasis should perhaps be on ensuring that the messages are effective (i.e., received as intended). However in general, both the scientist and the target of the communication must listen, understand, as well as speak.

Many academics find solace in science communication as an antidote to the challenges of higher education,
relishing the opportunity to step outside the confines of the ivory tower. As Dooley (2017) notes, when scientists engage in science communication, they should embrace their humanity and use emotions to communicate scientific concepts. This suggests that conversely, inside the ivory tower, academics may feel dehumanised (Wheaton, 2020). For example, academics report a sense of trepidation or fear around the completion of impact statements or when tick-box efficiency takes primacy over effectiveness (Chubb and Watermeyer, 2017; Chubb et.al., 2021). Engaging
with socio-economic and socio-cultural topics within science can help academics to get involved with new topics by developing an aspect of inspirational, or activating communication that can be regarded as a form of scholars' engagement (Jünger and Fähnrich, 2020). Our aim here is not to "police" the "right" objectives for academic science communications. As we highlight in the subsequent sections, our intention is to make science communicators and their (potential) funders reflect on the shadowlands of science communication. While there is
nothing inherently wrong in pursuing science communication as an antidote to higher education, we believe that it should not come at the cost of quality and rigor of the communication or the exploitation of communicators.





## 2.2 Poor quality and lack of rigor

Oftentimes, science communication tools do not work, and their failure can lead to enhanced disasters and loss of more lives (e.g., when miscommunicating about extreme weather events). In this section, we provide examples illustrating instances of poor quality and lack of rigor in science communication, with a focus on risk communication — a form of high-stakes science communication that occurs in challenging times. While this article primarily targets academia and academics, some examples are drawn from science communication outside 255 academia, intentionally so, since communication from government agencies (e.g., extreme weather and earthquake communication) often involves collaboration with university scientists.

For risk communication to be effective, it needs to capture and incorporate information about the local context in which the communication work is undertaken. Factors such as population characteristics (e.g., language, ethnicity, 260 and race), socioeconomic status, experience and exposure to a range of hazards, and access to and use of information and communications technologies influence the development and uptake of safety messages, and therefore, should be taken into consideration when designing communication outputs for decision making and advocacy in specific contexts. For example, the "Drop, Cover and Hold On" earthquake drills and campaigns considered how Californians behaved in past shakings (i.e., running outside, taking shelter in doorways, etc.), and 265 focused on the much greater likelihood of injury from non-structural hazards (i.e., falling or moving objects) rather than structural damage. To ensure its uptake, earthquake scientists and emergency managers worked closely with sociologists, artists and community participants to capture the regional context in the development and dissemination of disaster risk reduction messages.

Since 2008, the Shakeout campaign has gone global, with over 40 million participants registered worldwide for 2022. While there are good reasons to celebrate this, there are also reasons to be concerned. "Drop, Cover and Hold-on" may not be the safest action to take in highly vulnerable buildings that are small enough to exist safely (such as many of the buildings that collapsed during the 2005 Kashmir earthquake). Therefore, it is important to recognize that there is no single perfect safety message for any nation as each nation has its own customs, beliefs, 275 building, geology and capacities. A scientist who is not aware of local customs and deeply embedded beliefs should exercise caution when communicating safety messages with the public (Geohazards, 2018).

Hazard maps (in print and online) are another example of unidirectional communication output used by governmental and non-governmental agencies to communicate geohazard risks with the public. Despite their 280 widespread acceptance and use in hazard awareness campaigns and in decision making, their effectiveness in hazard communication has not been rigorously investigated. Setin et al. (2012) give examples of highly destructive earthquakes that occurred in areas shown by earthquake hazard maps to be relatively safe and call for rigorous and objective testing of hazard maps, and evaluation and clear communication of uncertainties with the users. Lack of basic elements of map reading skills is also identified as one of the key barriers to understanding earthquake-related 285 concepts amongst school students in Tajikistan (Mohadjer et al., 2021). While there are a few hazard map studies (e.g., Nave et al., 2010; Bell and Tobin, 2007; Crozier et al., 2006) exploring variables that influence people's map comprehension such as viewer perceptions of risk, risk area accuracy, preferences for map features, and misconceptions about visualizations, MacPherson-Krutsky et al. (2020) call for more research on assessing the degree to which different factors contribute to high map comprehension levels. Taken together, scientists as creators 290 of hazard maps need to engage in dialogue with a wide range of potential users to rigorously test and improve their communication products.



Good data visualization is a crucial means of communicating complex information in a clear and effective manner. Data visualization alongwith the representation of uncertainty plays a pivotal role in science communication, particularly when communicating complex information such as natural hazards or human-induced disasters. Poor data visualization can contribute to ineffective or subpar science communication, as highlighted by Padilla (2022), who discusses the challenges of conveying uncertainty through maps and emphasizes the need for effective visualization strategies to enhance comprehension of these uncertainties. Clear and accurate representation of uncertainty is relevant for many geoscientific challenges such as aftershock forecast maps (Schneider et al., 2022). The incorrect use of color in data visualization, as highlighted in Crameri et al. (2020), can also lead to misinterpretation of information.

Science communication can often be monodisciplinary. However, as pointed out above, collaboration between scientific disciplines (e.g., scientists studying specific hazards) and those assessing societal risk understanding (e.g., social or behavioral scientists) is essential for effective communication (Fischhoff and Scheufele, 2013). A recent example highlighting the lack of collaboration across relevant fields and science communicators, resulting in avoidable deaths, is related to the COVID-19 pandemic. In the early stages of the pandemic, debates arose regarding the modes of transmission of SARS-CoV-2, the virus that causes COVID-19. Morawska and Cao (2020), along with many aerosol scientists, argued that airborne transmission of the virus was a reality that should be acknowledged and addressed. They contended that the lack of attention to this primary mode of transmission in public health messaging led to a failure to implement adequate control measures, such as masking and improved indoor ventilation. Randall et al. (2021) provide a historical perspective on the transmission of respiratory infectious diseases and discuss how the lack of understanding of droplets and aerosols led to the undervaluation of the risk of airborne transmission for many respiratory infectious diseases, including COVID-19. The failure to recognize the role of airborne transmission in the spread of these diseases and the communication of incorrect science, including by the World Health Organization in the initial days of the pandemic, led to preventable illnesses and deaths.

These examples demonstrate how poor science communication and science communication systems (including absence of such systems) can have serious consequences and highlights the importance of accurate and clear communication of scientific information. Additionally, there have also been some (public) discussion of people mixing up public discussions on science and its results with discussions within science (e.g., climate change, COVID-19 vaccinations, etc.). Whilst scientists publish within scientific journals and on social media (e.g., Twitter, now known as X), traditional media, etc., "pseudo-scientists" only do the latter, but appear to be scientists to many people due to their loudness in social media and other media platforms. However, the public often cannot distinguish scientists and "pseudo-scientists" by these appearances and can think that there's no consensus where there is, or that critics are shut down. This is also an issue within science due to the widespread belief that uncertainties cannot be understood by decision-makers and that they cannot be incorporated into a binary yes/no decision-making process (Pappenberger and Beven, 2006). The information is therefore simplified to remove "unwanted" uncertainties. However, many decision-makers (e.g., for flood early warning) are well-versed in uncertainties, present in many other components of the forecast-based decision-making chain (Arnal et al., 2020; Budimir et al., 2020).

Despite communication being often at the heart of improved response throughout the disaster cycle (Golding et al., 2019), little attention has been given to the systematic evaluation of communication tools used or developed by scientists to inform and engage in dialogue with the public. These evaluations are important because effective communication, especially related to crises, has been shown to lead to more appropriate responses and the acceptance of more flexible hazard management strategies (Steelman and McCaffrey, 2013).



As discussed in the context of risk communication, a linear, unidirectional approach for increasing public awareness
does not always lead to action (Neil 1989; Tierney 1993; Fischhoff 1995; Sellnow et al., 2008). An effective
communication strategy takes into account the different ways people view risk, as well as their cultural and
socioeconomic context, all of which may affect how the risk is understood (Hooker and Capon, 2017; Cormick,
2019). Therefore, interaction and dialogue with those facing the risks can shed light on their risk perceptions and
how these relate to taking action (or the lack thereof) and provide essential insights into adapted and effective
communication strategies. These factors render the evaluation and comparison of communication difficult, as one
approach may be successful in a specific context and ineffective in other situations. While we focus on risk
communication in this section, the problems and discussions are relevant to all forms of science communication.

## 2.3 Exploitation of science communicators

### 2.3.1 The labor issue and exploitation of ECS and minoritized groups


There is general widespread pressure on all university-based scientists to communicate their research, this applies
a workload pressure to everybody, but impact differs according to time pressure, direction from funding bodies and
the provenance of academics (Hillier et al., 2019; Martinez-Conde, 2016). Anecdotally, at more senior levels,
mental health issues leading to breakdowns, marriage failure, and long-term stress are common symptoms which
can arise from emotional exhaustion and overwork (Hillier et al., 2019; Guidetti et al., 2020; Wheaton, 2020). The
hyper-competitive funding landscape for senior academics, according to Chubb and Watermeyer (2017), can rely
on the "research grants culture", or "game-playing" linked to inflated accounts of impact. There may also be a
tendency for more senior academics to displace the task of public engagement onto early career scientists (ECS),
or administrative staff – whether funded explicitly, or not, to do this (Watermeyer and Rowe, 2022; Pownall et al.,
2021). Despite these increased responsibilities for public outreach, ECS continue to have less established influence
or agency compared to their more senior colleagues. The tenure of ECS is predominated by short-term contracts
leading to reduced resilience, burnout or depression associated with academic precarity (Fowler, 2015; Wheaton,
2020; Hillier et al., 2019). Consequently, exploitation might have a different pathway and greater impact due to
perceived insecurities that are commensurate with the commencement of a career (Pownall et al., 2021).


ECS typically are encouraged to be involved with science communication as an activity crucial to developing the
next generation of scientists by improving scientific literacy within the public domain outside of academia (Kerr,
2021; Kompella et al., 2020). The motivations to engage with these activities can conversely be ascribed as
constraints as they are associated with the provision of public engagement activity that is identified as low-cost, or
a lesser value, and in many cases the mentoring of ECS by mid-career scientists is devalued (Hillier et al., 2019;
Kompella et al., 2020; Barrow and Grant, 2019). The potential for exploitation of their labor merits discussion and
can be contextualized within the broader concepts of pedagogic frailty, particularly as ECS constitute the most
numerous proportion of researchers in higher education (Kinchin and Francis, 2017; Lahiri-Roy et al., 2021;
Pownall et al., 2021). The impact of overwork as structural inequality endemic in academia arguably has
repercussions on the mental health of science communicators, indicating a clear link between the mental wellbeing
of academics and their perceptions of work demands. The prominence of research and public engagement demands
is recognized, which suggests the approach to these aspects of academia in terms of the potentially negative
consequences of exploitation and over-work, with evidence that these effects are most pronounced amongst
marginalized (minoritized) groups (Caltagirone et al., 2021; Wheaton, 2020; Guidetti et al., 2020; Barrow and
Grant, 2019; Hernandez et al., 2020).



The spectrum of marginalization occurs at an intersection of gender, race, caste, sexuality, physical ability, Global North vs Global South, and other identities and lived experiences which also influence how we see and study science and society (Canfield et al., 2020; Finlay et al., 2021; Lahiri-Roy et al., 2021). Geoscience, amongst all
STEM disciplines, has the lowest percentage of minoritized students and professionals which underlines this equity gap and the importance of the visibility of minoritized groups through public engagement is crucially important to breaking down stereotypes (Guertin et al., 2022; Weingart and Guenther, 2016). However, the assumption that minoritized groups must hold key responsibility to counter these affects through active, open and visible engagement pre-disposes marginalized groups to exploitation as communicators who are expected to provide
institutionally-led public engagement activity to counter prejudice and be equity-active (Barrow and Grant, 2019). Equity of marginalized groups in higher education is problematic and global discourse signifies a range of perspectives that can be adapted to fit cultural and social priorities. This needs to be tempered with the consideration of the ethics of equity in science communication, which undoubtedly shoulders a greater burden of responsibility to promote visibility of marginalized groups to marginalized science communicators (Barrow and
Grant, 2019; Lahiri-Roy et al., 2021; Caltagirone et al., 2021).

The "invisible" work of academia is pinpointed by the Social Sciences Feminist Network Research Interest Group (2017) as being a significant time drain on academics looking to develop their tenure and promotion. Furthermore, this invisible work can often be assigned to public engagement professionals which contributes to the
disproportionate demands on different roles that support science communication (Watermeyer and Rowe, 2022). The notion of invisible work is accepted as a norm within academia, particularly for women, which potentially pre-disposes an approach to science communication which can exploit public groups by calling on their "free" labor which exposes the unpalatable aspects of exploitation derived from in-kind contributions from unpaid co-producers (Vohland et al., 2021; Williams et al., 2020; Carter, 2020; Social Sciences Feminist Network Research Interest
Group, 2017). Support, in the form of mentoring, for women in STEM who return to work following a career break can be beneficial, conversely this can reinforce gender stereotyping whereby females are ascribed the roles of mentor under the misapprehension that they are perceived to be more motherly, caring, administrative or outreach orientated (Kompella et al., 2020; McKinnon and O'Connell, 2020). Given the minoritized position of women in STEM this undoubtedly denotes women as being a minoritized group at risk of exploitation, and this is starkly
evident within geoscience (McKinnon and O'Connell, 2020; Caltagirone et al., 2021; Hernandez et al., 2020).

### 2.3.2 Science communication jobs can kill scientific pursuits

The "Sagan Effect" refers to the risk that a science communicator may lose their scientific reputation among their
peers by simplifying concepts for a broader audience or being too visible (Chen et al., 2023). However, a survey of highly cited U.S. nano-scientists suggests that public communication, such as interactions with reporters and being mentioned on Twitter, can contribute to a scholar's scientific impact (Liang et al., 2014). Martinez-Conde (2016) argues that although most individuals who disseminate science to the public face no significant negative consequences and may even experience some benefits, there is a lack of recognition or rewards for their
communication efforts within institutional structures. Nevertheless, there are isolated cases where science communicators have experienced severe consequences. Furthermore, certain scientists from underrepresented groups may be at a higher risk of facing such negative consequences.

The impact of scientific research on society is frequently emphasized in academic job descriptions and promotion
criteria. According to Hillier et al. (2019), academic researchers may perceive engaging in knowledge exchange with industry as potentially detrimental to their career prospects due to time constraints. The study analyzes promotion criteria and job advert specifications, suggesting that for researchers to thrive, their impact work must





align with other demands on their time, such as research and teaching, which are currently deemed more crucial in academia. The relationship between impact work, research, and teaching might be more of an aspirational goal to
meet policy and funder expectations (Williams et al., 2020). Notably, higher-tier higher education institutions appear to have an advantage in securing research grants compared to lower-tier ones, highlighting an equity gap (Papatsiba and Cohen, 2020). Furthermore, while institutional policies often stress the importance of equity, it does not emerge as a significant factor in the promotion process for most academics (Barrow and Grant, 2019).

There are also some interesting parallels between our critique of the shadowlands of science communication to ongoing debates on on collaboration and coproduction. For example, Oliver et al. (2019) discuss the concept of coproduction in health research, which involves collaborating with stakeholders in the research process. They identify the costs associated with coproduced research and argue for a cautious approach to coproduction until more evidence is available on its impact and costs. Williams et al. (2020, p-1) respond "Oliver et al. stray too close
to 'the problem' of 'co-production' seeing only the dark side rather than what is casting the shadows. We warn against such a restricted view and argue for greater scrutiny of the structural factors that largely explain academia's failure to accommodate and promote the egalitarian and utilitarian potential of co-produced research." Similarly, in the case of science communication, even as we cast light on the shadowlands of science communication, we hope to also highlight the structural issues that cast these shadows.

## 445 3 Recommendations for (geo)science communication

The discussion in the previous section highlights the primary barriers for academics to carry out science communication sustainably and fairly, rather than reasons why they should not engage in science communication. The reasons to do science communication are still relevant even if institutional barriers make it hard to do so. In this section we discuss the specific recommendations for problems highlighted in Section 2 along with some best
practices.

### 3.1 Clarity and transparency in objectives and audience

Clarity and transparency in objectives and audience are critical components of effective science communication.
As Hutchins (2020) propose the following protocol to pursue an effective science communication: 1) Audience: Who will receive the communication and in what setting? 2) Purpose: What is the purpose of the communication? 3) Format: Will the communication product be oral, written, visual (or some combination) and what constraints does this format impose? 4) Significance: Communicating the story of your research. The audience is of utmost importance when customizing scientific communications and the success of the communication is ultimately
determined by the audience's response, making it the crucial metric for assessing whether the communication achieves its intended objective (Hutchins, 2020). Going a step further, Stewart and Hurth (2021) argue in favour of the more reflexive, participatory, and interdisciplinary "guide-and-co-create mode." From the perspective of this editorial, science communicators clarifying and being transparent about the objectives and audience of their science communication is also an effective way of countering the harmful and unclear objectives of science communication
(Section 2.1).

To tailor communications to specific audiences, it is necessary to create a profile of the audience, including their knowledge level and motivation for engaging in the communication. Additionally, it is important to consider the audience's cultural and social background, as these factors can impact how they receive and interpret information.
Similarly, the chosen language of science communication is also a tricky political question, as academia often





incentivizes use of English but local communities would benefit from local language(s). As Márquez and Porras (2020, p-5) note, "There is a language bias in the current global scientific landscape that leaves non-English speakers at a disadvantage and prevents them from actively participating in the scientific process both as scientists and citizens. Science's language bias extends beyond words printed in elite English-only journals. It manifests in how science is reported in mass and social media outlets, in the researchers represented in the media, and often in the lack of contact between communities and their local scientists"


Achieving effective science communication necessitates clarity and transparency in both objectives and audience engagement. By articulating the purpose of communication and grasping the characteristics and motivations of the audience, one can craft tailored communication products that effectively engage and inform. Moreover, highlighting the significance of research and fostering collaboration across diverse communities and languages can contribute to building a more inclusive and impactful scientific community. There is no singular approach to achieving this; rather, it requires the cultivation of expertise and competence within a community of practice—an objective at the core of *GC* for the geosciences community.



## 3.2 Training science communicators

While the importance of science communication is increasingly recognized and emphasized, many scientists do not receive any formal science communication training to develop the necessary skill sets. Science communication is often times done by scientists who are not adequately (or at all) trained in science communication (e.g., in visualization, social science, etc.), where ad hoc solutions are treated as substitutes for expertise in the sciences of communication (Fischhoff and Scheufele, 2013). While there are increasing amounts of informal training opportunities (e.g., academic conferences, talking to peers), to be effective, however, science communication must be part of an academic's formal training (Brownell et al., 2013). However, the opportunities at universities are very often irregular and informal. Examples include participation in community events on campus, science festivals (e.g., Pint of Science), presentation platforms (e.g., Three-Minute Thesis and TEDx), and media interviews.



Researchers training and development needs are summarised well in the Vitae Researcher Development Framework (RDF 2011). *Domain D* of the framework — Engagement, Influence, and Impact — covers the skills and knowledge needed for researchers to work with others and increase impact of the research. *Subdomain D2* Communication and dissemination and *Subdomain D3* Engagement and impact highlight the skills needed to excel in this area of research. Metcalfe (2019) reiterates that there is a divide between science communication models and theories used by science communication researchers and what happens in practice. There are three models described by Metcalfe (2019) — the Deficit model, the Dialogue model, and the Participatory model. Each with its own theories and set of necessary skills. However, their analysis of Australian science communication or engagement activities in 2012 discovered that most activities did not align their activity objectives with the underlying theory. More recently, Science Europe (2022) framework discusses a values based approach for organization of research, including for communication and dissemination of research, to facilitate 1) autonomy/freedom, 2) care and collegiality, 3) collaboration, 4) equality, diversity, and inclusion, 5) integrity and ethics, and 6) openness and transparency.




Communication skills form an integral part of researcher activities; however, these are often focused on dissemination of knowledge through outputs like research papers. What skills can be transferred to science communication from researcher development in general and what skills are specific to science communication? Kelp and Hubbard (2020) suggest that communication skills should be part of undergraduate education to establish a solid skills base. The Horizon 2020 project QUality and Effectiveness in Science and Technology communication




(QUEST) developed tools, recommendations and guidelines for communicators and practitioners (Costa et al., 2019). The QUEST WP4 summary report provides a comprehensive overview of science communication education across Europe. They recommend four key areas for science communication training: scientific knowledge, educational studies, social studies of science and communication studies. Offering a basic science communication training to all scientist as part of their development programme or studies is a key recommendation, with an element of broader societal context of the research, rather than skills development alone.

Some of the tools and approaches for science communication that should be taught are: conducting interviews, designing surveys, qualitatively/quantitatively analyzing interview/survey outputs, a basic understanding of ethics, designing serious games, storytelling, taking part in public debates, working with artists, art curators and art spaces. These tools should also target online communication and interaction (including on social media) and digital content creation (Bubela et al., 2009). Furthermore, training scientists in communication methods based on social science research and use techniques that involve the community in scientific issues will help challenge the deficit model and make science communication more effective (Simis et al., 2016)

More broadly speaking, we define the following three types of training needs:
- Most generic science communication is one-way dissemination of science and scientific work. In this case, training might be that of the journalist and the media world Storifying science.
- If the communication deals with informing the public about specifically socially contested ideas and issues, then the 'science of the public' (e.g., audience analysis, cognitive and social psychology) becomes an important training ground as a way of understanding two-way communication (dialogues; see Section 2.2) that allows better targeted messaging from scientists.
- A third training area is in three-way communication, where the aim is to provide science input into "social conversations about science" — deliberative fora such as citizen juries and assemblies or community-centered engagements — where the aim is to help empower people to use scientific knowledge for their own ends, which demands training in participatory and facilitative skills.

To improve science communication, Fähnrich et al. (2021) recommend that science communication programs and trainers should focus on developing students' mental models and perceptions of the changing societal framework in which science communication takes place through offering new insights, taking on new perspectives, supporting observations and reflection, and challenging world views. Incorporating training for geoscience students to learn and hone skills to communicate science to a wide audience into science curricula at an early stage (e.g., undergraduate level) can foster a better communication culture between scientific disciplines and the general public (Brownell et al., 2013).

As with scientific publishing, there is also a case to be made for "slow science communication" – prioritizing high quality over rapidness and quantity (Frith, 2020). Outcomes and impacts of science communication can also take time to bloom and hence may be hard to measure and demonstrate within the lifetime of most scientific projects.

### 3.3 Science communication as a professional activity

A large part of geoscience research is funded through government agencies around the world. These agencies are often funded by taxpayers, and as such, researchers have a responsibility to communicate their findings to the public. Unfortunately, few scientists around the world receive training in science communication aimed at the broader public. It should be noted that, in most parts of the world, scientists in academia do not receive training in teaching, even though they are expected to teach as part of their job responsibilities. In light of this, it is essential



that clear criteria for science communication be included as part of job requirements, with room for performance review and compensation. Science communication should also be incentivized for academic promotions. This
would be similar to how teaching is incentivized for promotions.

We need to emphasize the need to give science communication greater recognition, funding, and job opportunities. Additionally, Mulder et al. (2008) identified several steps for "bringing some order and appropriate recognition to the discipline of science communication: 1) Formation of a Register of Science Communication Programs; 2)
Recognition of a Core Framework; 3) Establish a Database of Resources for Teaching; 4) Establish a Major Prize for Science Communication". The American Geophysical Union reorganized in 2018 and elevated a marginal group (officially, a "Focus Group") "Science and Society" to "Section" status, making members of this section eligible for society-wide awards. There was still a lot of pushback on whether excellent communicators should become AGU Fellows and so a new Fellow-level award was created, the Ambassador Award. Similarly, the EGU
has the Katia and Maurice Krafft Award, which recognizes researchers who have developed and implemented innovative and inclusive methods for engaging with and communicating a geoscience topic or event with a diverse audience. Since 2015 it has also awarded Public Engagement Grants to celebrate and recognise excellent science communication in the Earth, planetary and space sciences. There is also this very journal (i.e., *Geoscience Communication*), which was in part set up to help reward researchers developing excellent science communication
and public engagement initiatives in the geosciences.

There is also a case made that not everyone can or should do science communication. Instead, we should support those who are good at it without making them suffer in the domain of their specialization. Irrespective of the stand of "scientists must participate in science communication" or "those who want to / are good at it should be
supported", we must be cautious not to fall into the trap of forcing minoritized groups to selectively carry out this invisible work. The Social Sciences Feminist Network Research Interest Group (2017) argue that in order to address the issue of invisible labor, we need to quantify and recognize the impact of this work, which is often overlooked or undervalued. We need to make the invisible visible in the case of science communication as well and give recognition to those who contribute their energies towards it.

In some countries science communication is mandatory for scientists to ensure career progress. For example, in Italy science communication is called the "third mission". At some institutions in the U.S., faculty do get positive annual salary review "points" for outreach. Some faculty even switched their appointment percentages to include outreach as part of their paid job, partly because there was a relatively easy venue (e.g., "Dinosaurs and Disasters
Day" at the adjacent natural history museum), and partly because of the way grants are structured in the U.S. The National Science Foundation requires outreach or some other clearly defined "Broader Impact" on grant proposals now. The PIs can do these "impacts" themselves or can hire education specialists or communications professionals to assist, but it must be in there. In Canada, where faculty performance is assessed based on annual reports, outreach (such as doing media interviews) is a sub-section on these reports, but it isn't clear to what extent it is valued
compared to other contributions such as graduating students or writing scientific papers.

While the push by some national funding agencies to promote science communication is welcome, science communication should also be considered a discipline in itself and requires efforts, as in any other field of research. Quite often, scientists believe that participating in events for the public is enough to assure good institutional
science communication. However, there are good reasons to not have all scientists participate in science communication. Incentivizing and training those scientists who are motivated to do so by a genuine interest may be a better approach. The scientific institution could take advantage of research groups in the field of science communication that are genuinely interested in identifying the most effective ways to involve the public in science.





Improving the assessment of scientific research output by funding agencies, academic institutions, and other entities has become an urgent necessity. In response, a group of scholarly journal editors and publishers convened at The American Society for Cell Biology's Annual Meeting in San Francisco in December 2012. Their objective was to create a set of recommendations, which is called the San Francisco Declaration on Research Assessment (DORA). DORA is now a global initiative that encompasses all academic disciplines (ASCB, 2012). It recognizes that

scholarly output extends beyond published journal articles and encompasses other items such as preprints, datasets, software, protocols, well-trained researchers, societal outcomes, and policy changes that result from research. In Canada, the Natural Sciences and Engineering Research Council of Canada (NSERC), in collaboration with four other Canadian research funding agencies, has endorsed this declaration.

In line with other scientific realms, science communication should establish clear norms regarding funders and partners to enhance transparency concerning potential vested interests of science communicators. This step ensures that the audience is informed of any external influences that may shape the narrative. Additionally, science communicators should clearly communicate their objectives with their audiences and obtain ethical clearances when relevant. Considering these aspects could help prevent deceptive campaigns, such as those with significant

environmental impacts. Furthermore, incorporating these dimensions into the practice of science communication fosters a more transparent and ethically sound landscape, thereby enhancing the credibility and integrity of the field.

## 4 Final thoughts


Science communication is a vital aspect of the scientific enterprise, and it is our responsibility to communicate scientific concepts and discoveries to non-specialist audiences. However, as we throw a light on the shadowlands of science communication, we also want to clarify that we do not want to discourage scientists from talking to kids, to teachers, to the public, and especially to legislators. There is a spectrum of science communication and science

communicators within and outside of the academy (Illingworth, 2023), and all of it plays an important role — even if not "professionalized". However, we must make clear criteria for science communication as part of job requirements, incentivize science communication for academic promotions, and support those who are good at it without making them suffer in the domain of their specialization. We must also ensure that the impact of science communication is visible and valued.


While the case in favor of science communication has garnered significant attention in recent years, it is equally crucial to contemplate why not all academics should be compelled to engage in science communication. This consideration becomes especially pertinent within the context of an already exploitative environment, namely academia. Science communication, when undertaken indiscriminately, may not adhere to the same standards of

honesty and rigor expected from either scientists or journalists. Additionally, it is impractical and inefficient to expect every academic to excel in all sub-specializations, encompassing research, teaching, enterprise, communication, and more.

Instead, a more equitable approach entails recognizing the intrinsic value of specialized expertise in the field of

science communication and providing unwavering support to dedicated professionals in this domain, while safeguarding against exploitation and potential detriment to their long-term careers. By adopting this approach, we can contribute to a more transparent and responsible landscape within the realm of geoscience communication, effectively addressing concerns related to exploitation and the invisibilization of the invaluable contributions made by science communicators. Such efforts will ultimately preserve the credibility and efficacy of science



communication, facilitating the public's enhanced understanding of scientific concepts, and thereby benefiting
science, scientists, and society as a whole.

This editorial is based on a review of the literature and our own experience, but having a more in-depth analysis
through surveys distributed at conferences, etc. could help identify which issues are truly pervasive, and perhaps
even highlight new issues we hadn't thought about. We also hope that the open review process for this manuscript,
including community comments, will help us improve and add further perspectives to this editorial.

**Data availability.** No primary data sets were used in producing this article.

**Author contributions.** Conceptualization and methodology: All authors; Project administration: SG; Writing –
original draft: IS, JH, KP, KvE, LA, LB, SG, SM, TL; Writing – review & editing: All authors.

**Competing interests.** HR, IS, LA, SG, TL are editors of *GC*; JH, KvE, SM are executive editors of *GC*; SI is the
chief-executive editor of *GC*.

**Ethical statement.** This editorial reflects the authors' views and does not involve sensitive data or human
participants; as a result, no ethics approval or informed consent was sought.

**Acknowledgements.** We would like *GC* editors Leslie Almberg, Mary Anne Holmes, Mathew Stiller-Reeve, and
Katharine Welsh for participating in initial discussions about this article. We would also like to thank Raymond
Spiteri for his intellectual guidance. We would also like to express our gratitude for the numerous informal
discussions we have had with scientist and science communicator colleagues over the years. These exchanges have
not only served as a source of inspiration but have also significantly contributed to the content of this editorial.

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
