# Peer review of "Editorial: The shadowlands of (geo)science communication in academia — definitions, problems, and possible solutions"

_EGUsphere, 2023_

## Author Comment (AC3)

**Community comment 1:**

We want to thank Dr. Heather Doran for her comments.

My comment relates to the role of science communication facilitators in this landscape. It isn't clear (until later in the article) that you only discuss the role of scientist - communicators in this overview. Universities employ public engagement professionals, science writers, events organisers and or outreach organisers who take on some of the burdens of science communication to support, train and facilitate communication from scientists. It would be great to see this being explored in more detail in the discussion. You could explore some of the resources from the National Coordinating Centre for Public Engagement in the UK for further information about what these engagement professionals do in the academic arena. In addition they have undertaken activities that relate to recognition of communication and engagement within the role of research. There's a very interesting discussion about who delivers training in this area and how this might adhere to a set of principles rather than it being based only on an individuals experience. Depending on how much that individual is connected with the wider discussions in science communication and/or networks such as the Public Communication of Science and Technology Network (PCST) they may or may not be aware of some of the complexities in this space.

This is a good point, and we will clarify that our focus in this editorial is indeed on scientists in academia. That said, we will also include a discussion on science communication facilitators in the revised manuscript (e.g., in Section 3.3 'Science communication as a professional activity').

More information about the relationship between scientists and citizen scientists in this overview of how science communication may take place. I don't think this dynamic is fully reflected in the overview as it stands - this could be explored through the Science Shops network and/or the Impetus Citizen Science project.

Considering our focus on the 'shadowlands,' particularly within the context of geoscience communicators (scientists), this point may fall outside the scope of this editorial. However, we will briefly address it in section 3.3. Furthermore, we plan to elaborate on this aspect in the Introduction.

---

## Author Response (AR1)

Blue color is used for response to comments

"Maroon color is used to quote text in manuscript"

We want to thank our reviewers, the individuals who provided community comments on the manuscript, and the editor for reviewing this manuscript.

**Referee 1**

We want to thank the reviewer for their comments.

Thank you for the opportunity to read your work "The shadowlands of science communication in academia — definitions, problems, and possible solutions." Working in the science communication space, I agree that the "shadowlands" is a great way to represent the elements of the field that are so central to our work and yet so ambiguous or unclear for those trying to navigate research and meaning in science communication. I think that this piece will shed light on the unwritten and often muddy spaces that dictate a lot of our work, but are often not explicitly taught, accounted for, or made clear. Thinking through shadowland spaces will be helpful for both current and future science communicators, in geoscience, but also more broadly.

I appreciated the introductory material that offers a brief history of where science communication has been and where it is going next. Something that I found myself wondering about was the connection between the science communication definitions offered, the specific Geoscience Communication definitional spectrum, and the taxonomy that followed. I thought that outlining the elements of geoscience areas was helpful, but thought there could be a bit more clarity situating this very specific definitional framework against the larger definitions of science communication, particularly since the section that follows offers another set of categorizations. I wasn't always sure how to connect the three different definitional frameworks. Once the shadowlands section began, I was more clear on the trajectory.

We have now condensed and streamline Section 1.1 and 1.2 to address these points (similar suggestion by Referee 2).

Figure 1 offered a really clear overview of the project and goals of the piece in terms of framing science communication. One question I had for further consideration is: what are the distinctions between training science communicators and viewing science communication as a valued professional activity? There seems to be overlap in these two solutions, but perhaps an example or a bit more detail could differentiate the two.

"Training science communicators" pertains to the individuals engaged in communication efforts, whereas "viewing science communication as a valued professional activity" focuses on the broader institutional recognition and support for science communication within academic circles. This distinction is elaborated upon extensively, with accompanying examples, in Sections 3.2 and 3.3, respectively. We have now used the figure caption to redirect readers to the individual sections where they can find information.

I really appreciated the discussion about responsible use of uncertainties on page 9, and wondered if it might be worth mentioning that public audiences are also equipped to handle uncertainties as decision-makers are (unless decision-makers in this case was meant to represent both public and expert audiences). Overall, I thought the discussion about unidirectional risk communication was really timely and important.

We have reviewed the text in Section 2.2 to draw a distinction between the public audience (individuals making personal decisions) and decision-makers (making decisions for the community/public).

This issue also persists within the scientific community, partly due to the belief that uncertainties cannot be understood by decision-makers and the public, and therefore cannot be incorporated into a binary yes/no decision-making process (Pappenberger and vavBeven, 2006). As a result, information is often simplified to remove 'unwanted' uncertainties. However, many decision-makers (e.g., those involved in flood early warning) are well-versed in handling uncertainties, as these are present in many other components of the forecast-based decision-making chain (Arnal et al., 2020; Budimir et al., 2020). Additionally, public audiences can also engage with uncertainties when communicated effectively (van der Bles et al., 2020).

In terms of the first recommendation regarding Clarity and transparency, I think the term clarity could be explored in a bit further detail. Clarity can mean a lot of different things depending on context, which might be worth discussing. I do think, though, that the focus on audience in this section is key and does a great job of highlighting that cultural and social backgrounds impact audience interpretation and interaction with the information.

We have provided definitions of "clarity" and "transparency" in this context at the start of the section. As suggested, we have reviewed the use of "clarity" throughout Section 3.1.

"Clarity in science communication pertains to the accurate and straightforward transmission of information, ensuring that the intended message is effectively conveyed and understood by the audience without confusion. Transparency, meanwhile, involves being forthright about the goals, context, and any underlying biases or constraints influencing the communication. Together, clarity and transparency are essential for fostering trust and understanding between scientists and their audiences. Clarity and transparency are critical components of effective science communication. Hutchins (2020) proposes the following protocol to pursue effective science communication:

1) Audience: Who will receive the communication and in what setting?
2) Purpose: What is the purpose of the communication?
3) Format: Will the communication product be oral, written, visual (or some combination), and what constraints does this format impose?
4) Significance: What is the significance of the research for this audience?
5) Get feedback and revise

Understanding the audience and the purpose of the science communication is paramount when tailoring messages to ensure effective engagement. The success of communication is ultimately gauged by the audience's response, making it a critical metric for assessing whether the communication achieves its intended objective. Clarity is context-dependent and involves more than simply simplifying complex information; it requires careful consideration of language, tone, and framing to align the message with the audience's needs. For example, in a technical report aimed at experts, clarity may be achieved through precision and specificity, whereas in public outreach, clarity may necessitate simplicity and engagement."

The authors did a great job of discussing specific organizations that could modify their practices with regards to science communication to offer a way forward, and the level of detail constitutes a great first step towards these ends. If anything, some information about first steps that readers could take towards implementing or advocating for these goals could make these goals more actionable for those in the field who may not be in direct positions of power to influence these decisions. Overall, though, I thought these ideas were great, and geoscience communicators can really benefit from considering these alternative ways of valuing and incentivizing science communication in our diversity of roles.

We have added a paragraph in Section 4 that details the initial steps readers can take. These steps are distinct from the broader changes needed within the academic ecosystem.

"To make the broader goals discussed in this editorial more actionable for those not in direct positions of power, readers can take several initial steps:

1. Advocate for inclusive training opportunities: Encourage the integration of science communication training into professional development and academic curricula. Ensure that such training addresses diverse perspectives and includes underrepresented groups to promote equity in science communication.
2. Promote and share best practices: Share and implement effective science communication strategies within your institution and professional network. Prioritize practices that respect and value the contributions of all communicators, and address any systemic biases that might affect their involvement.
3. Support and mentor colleagues: Provide resources, constructive feedback, and mentorship to early-career colleagues interested in science communication, while recognizing that mentoring is valuable at all career stages. Foster a collaborative environment where early-career scientists can receive guidance and where more experienced colleagues can benefit from fresh perspectives and feedback. Additionally, nominate collaborators, colleagues, or employees who demonstrate excellent work in geoscience communication for recognition, awards, and prizes within their institutes or at national and international levels (e.g., conferences).
4. Engage in equitable dialogue: Initiate and participate in discussions about the importance and value of science communication. Advocate for fair recognition and compensation for science communicators, and work to build broader support within your community while being mindful of the different challenges faced by underrepresented groups."

Other considerations:

- In line 211, "some scientific discipline" should read "scientific disciplines"
- In line 272, should "action" be "actions" based on the multiple recommendations (drop, cover, hold on)?
- There appear to be a few extra spaces in line 392

We have made the three corrections listed above.

**Referee 2**

We want to thank Dr. Robyn Pickering for her comments.

Gani et al present a thoughtful, well researched and compelling editorial on what they term the 'shadowlands' of science communication. They clearly present the value of science communication and identify the issues pushing this into the shadowlands. They then offer some interesting perspectives and clear recommendations. This is a valuable piece and I enjoyed reading it very much and am delighted to be in a position to offer some comments.

My main and central comment is on the use of language and the distinction between 'science' (implying the entire arc of all things scientific) and 'geoscience' (more focused on the earth and earth processes). The use of both terms and both meanings belongs in this piece but there are many times where narrowing the discussion and scope to 'geosciences' I believe is appropriate. This is a Geoscience journal, from a geoscience society, written by a group of geoscientists, for an audience predominantly of geoscientists! I don't wish to labour this point, but I think the addition of 'geo' to many of the instances in which 'science' is mentioned will focus and strengthen the arguments and piece in general. There are generic points about science communication but the examples and recommendations, especially related to hazards, are more geoscience. My recommendation is that the authors critically assess almost every mention of 'science' and see if replacing this with 'geoscience' would work better. For example section 3.3 could be 'Geoscience communication as a professional activity'.

We agree with the reviewer and have: 1) focused the discussion on geoscience communication as much as possible, especially in Section 3 (now titled *"Recommendations for (geo)science communication"*), and 2) thoroughly reviewed the entire text to reassess the use of "science" versus "geoscience."

I am not sure if section 1.2 is necessary - it's a nice literature review but in my view does not add much to the paper and could either be condensed into the introduction or left out. This will also make the piece shorter and more focused.

We have condensed and streamline Sections 1.1 and 1.2 to address these points (similar suggestion by Referee 1).

In section 2, the term 'shadowlands' is discussed in more detail. My sense is these authors are introducing this term for the first time? If yes, maybe add some text saying 'in our opinion' or 'from our perspective'. This is such a useful phrase and way of looking at improving geoscience communication, I think the authors should take credit for it!

We have incorporated the change suggested towards the end of Section 1.

"In line with the core purpose of GC, in this editorial we highlight systemic issues ingrained in science communication, especially as it relates to the geosciences and geoscientists in academia. We refer to these issues as "shadowlands" hereafter."

Figure 1 is really clear and useful.

Thank you

In section 2.3.1, I wonder if the authors would like to be even clearer and call out the Whiteness of geoscience - rather than just say how low the percentage of minoritized groups. There are two articles which clearly articulate the Whiteness of geosciences: Dutt, 2020, Nature Geoscience and Berhe et

al., 2022, Nature Geoscience. From my perspective, this section can be strengthened by being clearer - geosciences does not just have a low percentage of everyone else, our field is predominantly White and this Whiteness carries a lot of privilege.

This is an excellent suggestion. We will incorporate it and are grateful for the references. (see next point for updated text)

Following on from this point, in the next paragraph, there is a very well written argument about the 'invisible labour' done mainly by women. Again, I wonder if this section can be even more specific, rather than just saying this labour is done mainly by women, say that male privilege shields many geoscientists from feeling pressured or obliged to undertake this labour. Then intersect this with race, and we have White male privilege vs minoritized women burdened with additional and invisible labour. A recommendation out of making this discussion more explicit could be that better geoscience communication needs to be more representative, which in this case requires broader participation.

We have incorporated this recommendation in our revised manuscript. The following is the updated text pertaining to both this and the previous point.

"The spectrum of marginalization occurs at an intersection of gender, race, caste, sexuality, physical ability, Global North vs Global South, and other identities and lived experiences which also influence how we see and study science and society (Canfield et al., 2020; Finlay et al., 2021; Lahiri-Roy et al., 2021). Geoscience, amongst all STEM disciplines, has the lowest percentage of minoritized students and professionals which underlines this equity gap. The field is predominantly White, carrying substantial privilege (Berhe et al., 2022; Dutt, 2020). The visibility of minoritized groups through public engagement is crucially important to breaking down stereotypes (Weingart and Guenther, 2016; Guertin et al., 2022). However, the assumption that minoritized groups must hold key responsibility to counter these affects through active, open and visible engagement pre-disposes marginalized groups to exploitation as communicators who are expected to provide institutionally-led public engagement activity to counter prejudice and be equity-active (Barrow and Grant, 2019). Equity of marginalized groups in higher education is problematic and global discourse signifies a range of perspectives that can be adapted to fit cultural and social priorities. This needs to be tempered with the consideration of the ethics of equity in science communication, which undoubtedly shoulders a greater burden of responsibility to promote visibility of marginalized groups to marginalized science communicators (Barrow and Grant, 2019; Caltagirone et al., 2021; Lahiri-Roy et al., 2021).

The "invisible" work of academia is highlighted by the Social Sciences Feminist Network Research Interest Group (2017) as being a significant time drain on academics looking to develop their tenure and promotion. This invisible work can often be assigned to public engagement professionals, contributing to disproportionate demands on different roles that support science communication (Watermeyer and Rowe, 2022). The notion of invisible work is accepted as a norm within academia, particularly for women, which may lead to the exploitation of public groups by relying on their "free" labor, revealing unpalatable aspects of exploitation derived from in-kind contributions from unpaid co-producers (Social Sciences Feminist Network Research Interest Group, 2017; Carter, 2020; Williams et al., 2020; Vohland et al., 2021). Support in the form of mentoring for women in STEM returning to work following a career break can be beneficial; conversely, it can also reinforce gender stereotyping when females are assigned mentoring roles under the misapprehension that they are perceived as more "motherly," caring, administrative, or outreach orientated (Kompella et al., 2020; McKinnon and O'Connell, 2020). This dynamic underscores the interplay of male privilege, particularly White male privilege, which shields many geoscientists from the pressures and obligations of invisible labor, while minoritized women are burdened with additional and invisible work (Hernandez et al., 2020; Caltagirone et al., 2021)."

In summary, this is a thoughtful, well written, timely piece which will generate further discussion, as well as recording where we are right now in understanding and bettering geoscience communication. I look forward to seeing the final version published.

**Community comment 1:**

We want to thank Dr. Heather Doran for her comments.

My comment relates to the role of science communication facilitators in this landscape. It isn't clear (until later in the article) that you only discuss the role of scientist - communicators in this overview. Universities employ public engagement professionals, science writers, events organisers and or outreach organisers who take on some of the burdens of science communication to support, train and facilitate communication from scientists. It would be great to see this being explored in more detail in the discussion. You could explore some of the resources from the National Coordinating Centre for Public Engagement in the UK for further information about what these engagement professionals do in the academic arena. In addition they have undertaken activities that relate to recognition of communication and engagement within the role of research. There's a very interesting discussion about who delivers training in this area and how this might adhere to a set of principles rather than it being based only on an individuals experience. Depending on how much that individual is connected with the wider discussions in science communication and/or networks such as the Public Communication of Science and Technology Network (PCST) they may or may not be aware of some of the complexities in this space.

This is a good point, and have clarified that our focus in this editorial is indeed on scientists in academia. That said, we have included a discussion on science communication facilitators in the revised manuscript (Section 3.3: *"Recognize science communication as a valued professional activity"*).

"In addition to scientists, some universities nowadays also employ public engagement professionals, science writers, events organizers, and outreach coordinators who support and facilitate communication from scientists. These professionals play a crucial role in easing the communication burden on scientists and ensuring effective public engagement. Their contributions should also be recognized and supported within the academic structure. However, it is important to restate that our focus in this article remains on geoscientists engaging in geoscience communication."

More information about the relationship between scientists and citizen scientists in this overview of how science communication may take place. I don't think this dynamic is fully reflected in the overview as it stands - this could be explored through the Science Shops network and/or the Impetus Citizen Science project.

Given our focus on the 'shadowlands,' particularly within the context of geoscience communicators (scientists), this point falls outside the scope of this editorial. We have clarified our scope throughout the article.

**Community comment 2:**

We want to thank Dr. David Crookall for taking the time to review this manuscript and for providing valuable comments.

I enjoyed reading the ms; it helped me greatly to understand much about the areas that need work in geosci comm. I am sure that it will help the whole community of geoscience – not just geosci comm.

Here are a few thoughts that might be useful. I hesitated about whether or not to offer these thoughts, fearing that they may be taken as intended, that is, as a small bowl of ideas, among which you may pick and choose, with the hope that they might be useful as you revise your ms.

Your mention of the deficit model reminds me of a number of works:

For example, C P Snow's two cultures, back in 1959, and book in 1963, argued that our society had divided into two mutually 'inunderstandable' groups.

The ground-breaking study by Baratz & Bafratz, 1970, "Early Childhood Intervention: The Social Science Base of Institutional Racism" shines a light on the idea of educational deficit. They propose a different model: "The cultural difference model is presented as a viable alternative to the existing genetic inferiority and social pathology models". I must say that I still encounter thickly-veiled attitudes of that kind in academia.

We appreciate the historical context provided regarding discussions surrounding the 'deficit model'. Delving deeply into the "deficit" and other models is beyond the scope of this manuscript, particularly considering that we will be shortening this section in accordance with reviewer comments. However, we do begin to examine the importance of considering cultural differences for effective geoscience communication in the manuscript. For example, in Section 2.2, we state, "An effective communication strategy takes into account the different ways people view risk, as well as their cultural and socioeconomic context, all of which may affect how the risk is understood "

Somewhat related to the two cultures is the vibrant discipline of cross-cultural (or intercultural) comm and training. One might say that the lay public and scientific groups tend to live in two fairly distinct cultures, each with their own beliefs, values and behaviours. Stella Ting Toomey and Young Yun Kim – and their colleagues – have done some marvellous research in the area. They may sometimes talk with each other, but the cultural gap remains wide. We geoscientists wishing to visit a lay culture need to learn much before we travel there. One unwritten rule of cross-cultural travel is that visitors must adapt to the host.

Sometimes, I wonder whether the general public and scientific groups do not get caught up unawares in the quagmire of sociocognitive processes, such as social identity, categorization, intergroup behaviour and language – see, the work by Henri Tajfel, Howie Giles, John Taylor, Miles Hewstone and others. In particular, I would suggest that you include the social psychology of language (Giles), which is a powerful force in intergroup comm and understanding, especially as comm, sci comm and outreach inevitably use language in massive doses. (British people know (or should know) the force of accent and speaking style in comm and intergroup behviour.)

Your potential solutions (Fig 1) are excellent, but my fear is that, without some consideration of the above dimensions, they will remain relatively superficial, and lead to results that are disappointing in the long run. I suggest that your solutions should include derived from the intercultural and intergroup language behaviour domains.

One way to bridge the divide is citizen science, esp citizen sci that is done in close collaboration with professional scientists. Both groups can be trained in intercultural comm and then encouraged to work together on an equal footing, not just in doing fieldwork, but also in publishing. The geosciences offer great opportunities for that.

While the above comments and literature appear intriguing and relevant to the discussion, we do not believe that we, as geoscientists engaged in and discussing geoscience communication, are fully equipped to address these points. This aligns with Reviewer-2's suggestion to confine the discussion to geosciences, and it's important to reiterate that our focus remains on geoscientists in academia. Nevertheless, your points indicate the necessity of explicitly stating our limitations to the reader. We have address these limitations in Section 4, 'Final Thoughts'.

"This editorial is based on a review of the literature and our own experiences, with a focus on geoscience communication. It is not a comprehensive review of the entire field of science communication. The challenges discussed are primarily informed by contexts in the Global North; however, similar shadowlands of science communication likely exist in other regions, influenced by

factors such as race, gender, ethnicity, religion, language, and caste. An in-depth analysis through surveys or additional research could reveal more pervasive issues and highlight new challenges. We hope the insights shared here inspire and inform efforts to enhance fair science communication across diverse contexts and disciplines."

I would not agree completely with the statement: "leaves non-English speakers at a disadvantage and prevents them from actively participating". Many French, German, Spanish, etc. scientists are in fact at an advantage (assuming of course that they are at ease with what is called 'international English'). Have you noticed that at some conferences, the non-English people seem to gather easily among themselves without natives? I can tell you a convincing, personal story sometime about that sometime.

We have directly quoted Márquez and Porras (2020, p-5) who are discussing this specifically in the context of 'non-English speakers' who are not at an ease with English.

Also, I think that some discussion about the fundamentals or theories of comm might help, esp as comm is so diverse – in its nature and in the scholarship and research about it. For example, in l.534, you talk about one-way comm. This smacks of the old Shannon & Weaver model of comm. It might be worth bringing in constructivist approaches, eg, Berger & Luckmann's classic work on meaning making and legitimization.

Thank you for the comment, but this is beyond the scope of this editorial.

I wonder whether it would help the reader if you developed some summary tables at various junctures. Also, diagrams illustrating in visual form some of your concepts would be most welcome.

We have included a new Figure (Figure 2: Taxonomy and goals of science communication based on literature).

It would be marvellous if you were able to expand on your training needs (l.532), and even provide examples of training methods (such as simulation/gaming and debriefing) that match those needs.

We have now expanded on the training needs (Section 3.2) as suggested by the reviewer.

"Some of the tools and approaches for science communication that should be taught are: conducting interviews, designing surveys, qualitatively/quantitatively analyzing interview/survey outputs, a basic understanding of ethics, designing serious games, storytelling, taking part in public debates, and working with artists, art curators, and art spaces. These tools should also target online communication and interaction (including on social media) and digital content creation (Bubela et al., 2009). Furthermore, training scientists in communication methods based on social science research and techniques that involve the community in scientific issues will help challenge the deficit model and make science communication more effective (Simis et al., 2016).

More broadly speaking, we define the following three types of training needs:

1. One-way communication: Training for one-way dissemination of science and scientific work focuses on the skills used by journalists and media professionals to present science in a compelling narrative form. For example, writing a news article about a recent scientific discovery or creating a documentary that explains complex scientific concepts to a general audience.
2. Two-way communication: When the communication aims to inform the public about socially contested ideas and issues (e.g., climate change, vaccination, genetically modified organisms), understanding the 'science of the public'—such as audience analysis and cognitive and social

psychology—becomes crucial. This type of training helps scientists engage in dialogues that allow for more targeted and effective messaging.

3. Three-way communication: The goal here is to contribute scientific input to broader "social conversations about science," such as those in deliberative forums like citizen juries, assemblies, or community-centered engagements. This approach empowers individuals to use scientific knowledge for their own purposes, requiring training in participatory and facilitative skills."

Some mechanical things :

l.26 -- which: does that refer to shadowlands or to practices or to something else? is the subordinate defining? (then use that) or non-defining? (then use , which)

Fixed

"These shadowlands are spaces, aspects, and practices of science communication that are not clearly defined and may be harmful with respect to the science being communicated or for the science communicators themselves."

l.212 -- 's' missing

Fixed

l.455 -- it would help the reader if your numbered sentences were placed vertically (as in bulleted items)

We have incorporated the suggestion.

"Hutchins (2020) propose the following protocol to pursue an effective science communication:

1) Audience: Who will receive the communication and in what setting?

2) Purpose: What is the purpose of the communication?

3) Format: Will the communication product be oral, written, visual (or some combination) and what constraints does this format impose?

4) Significance: What is the significance of the research for this audience?

5) Get feedback and revise"